# Dye-Encapsulated Metal–Organic Frameworks for the Multi-Parameter Detection of Temperature

**DOI:** 10.3390/molecules28020729

**Published:** 2023-01-11

**Authors:** Yating Wan, Yanping Li, Dan Yue

**Affiliations:** 1Intelligent Manufacturing College, Hangzhou Polytechnic, Hangzhou 311402, China; 2Shanghai Institute of Laser Plasma, China Academy of Engineering Physics, Shanghai 201800, China; 3College of Information Science and Engineering, Changsha Normal University, Changsha 410100, China; 4Henan International Joint Laboratory of Rare Earth Composite Materials, College of Material Engineering, Henan University of Engineering, Zhengzhou 451191, China

**Keywords:** metal–organic frameworks, thermometer, wide range, multi-parameter detection, quantum yield, highly sensitive

## Abstract

Temperature is an important physical parameter and plays a significant role in scientific research, the detection of which cannot be too crucial to study. In order to reduce the interference of the external environment on the detection of temperature and improve the accuracy of the detection results, a multi-parameter detection method using several optical signals was proposed. Here, a novel porous metal–organic framework (MOF), Zn-CYMPN, was synthesized and structurally characterized. Then, fluorescent organic dyes, either DPEE or DPEM, were encapsulated into the pores of Zn-CYMPN independently. The successful synthesis of the composites Zn-CYMPN⊃DPEE or Zn-CYMPN⊃DPEM could easily introduce other fluorescent centers into the original material and made it more convenient to realize multi-parameter temperature detection. More specifically, when the temperature changed, the maximum fluorescent emission wavelength (*W*) and the maximum optical intensity (*I*) of the Zn-CYMPN⊃DPEE/DPEM both showed good linear responses with temperature over a wide range, indicating that the composites were highly sensitive thermometers with multi-parameter temperature readouts. In addition, the quantum efficiency and thermal stability of the organic dyes, which bother every researcher, were improved as well.

## 1. Introduction

Temperature is a very important parameter in the fields of daily life, industrial production and scientific research, so how to achieve accurate temperature detection is a promising research direction [1,2]. At present, commonly used thermometers, such as thermocouples, are not suitable for use in harsh environments, in strong electromagnetic fields or with high-resolution requirements [3,4]. Fluorescence sensors are a feasible solution to overcome this problem, especially since a series of optical materials with excellent properties and various fluorescent colors have been developed to detect temperature. They are one of the most noteworthy measurement methods due to their non-invasive nature, fast response and ability to work in intensive environments with a high spatial and temperature resolution [5,6,7]. So far, most of the reports about temperature detection by fluorescence-based thermometers are based on changes in the fluorescence intensity, lifetime and maximum emission wavelength as well as the fluorescence intensity ratio, separately [8,9,10]. However, it is also a feasible method to use a variety of optical signals for temperature detection. Compared to those single-parameter methods, these multi-parameter thermometers, which use multiple optical signals simultaneously for temperature detection, show higher accuracy. Multimode sensing was realized by encapsulating 4-Mu and Flu into the pores of a nanoscale metal–organic framework (MOF), ZIF-8 [11]. However, due to the fact that relevant material design is more difficult for researchers, specifically, the selection of various materials and a combinational design of different optical signals to realize multi-parameter detection, there have been few studies on it.

At present, the research on fluorescent sensing materials has been more and more extensive and in-depth [12]. Dyes, as a big classification of fluorescent sensing materials, tend to have excellent optical properties with high quantum yields in dilute solutions, while their powders present a significant decrease in fluorescent properties due to strong aggregation-caused quenching (ACQ) [13,14,15]. However, solutions cannot be used as detectors at high temperature due to their volatility and the difficulty in handling them relative to solid samples. Therefore, it is practicable to improve the applicability and quantum yields of dyes by assembling both them and porous materials, such as porous silicon, zeolite and layered inorganic salt, to form composites [16,17]. Host–guest materials have been applied to many fields such as fluorescent materials, lasers and nonlinear optics [18,19,20,21]. However, inorganic porous materials have many crystal defects, and the shape and size of their internal pores are hard to design [22]. MOFs are crystalline materials synthesized by the self-assembly of metal ions and organic ligands [23]. Benefiting from their exceptional porosity and structural diversity, dyes can be reasonably encapsulated into their pores for functionalized applications. In addition, the negligible leak-out can eventually generate temperature-dependent fluorescent emissions [24,25,26]. To date, some MOF-based fluorescent thermometers have been reported, but the majority of them have been fabricated based on a mixed-lanthanide MOF strategy [9,27]. Since the emission of the lanthanide metal is a characteristic optical phenomenon of these materials, the emission peak hardly moves with the change in temperature, which makes the thermometric parameter be limited to the fluorescence intensity. In order to achieve a multi-parameter temperature probe design, and due to their rich variety and their wide spectral range ranging from ultraviolet to near infrared, fluorescent dyes can be utilized for realizing fluorescent detection with multiple parameters [28]. MOF⊃dye systems, which utilize the maximum emission wavelength and optical intensity of dyes to simultaneously detect temperature and reduce the ACQ of dyes, are a promising approach to develop multi-parameter fluorescent thermometers [29].

Herein, a novel MOF, namely Zn-CYMPN (H_6_CYMPN = 6,6’-[[2,2-bis[[(6-carboxy-2-naphthalenyl)oxy]methyl]-1,3-propanediyl]bis(oxy)]bis-2-naphthalenecarboxylic acid), with multiple channels was synthesized. The longer side chains of H_6_CYMPN may interpenetrate with each other, which will form smaller channels, helping the guest molecules to bind more firmly. Then, the fluorescent dyes 4-[2-[4-(dimethylamino)phenyl]ethenyl]-1-ethylpyridinium (DPEE) and 4-[2-[4(diethylamino)phenyl]ethenyl]-1-methylpyridinium (DPEM) were encapsulated into the pores of the Zn-CYMPN to yield the MOFs⊃dye composites Zn-CYMPN⊃DPEE/DPEM. The resultant dye-encapsulated composites Zn-CYMPN⊃DPEE and Zn-CYMPN⊃DPEM could achieve multi-parameter fluorescent temperature readouts in a wide range with a high spatial resolution and good thermostability.

## 2. Results and Discussion

### 2.1. Structure of Zn-CYMPN

Fine granular crystals Zn-CYMPN were yielded by the solvothermal reaction of H_6_CYMPN and Zn(NO_3_)_2_·6H_2_O in mixed solvents of DMF and water. Single-crystal X-ray diffraction analysis revealed that the Zn-CYMPN crystallizes in the monoclinic space group *I2* (Appendix A) with the structural formula Zn_3_(C_49_H_32_O_12_)_2_·(H_2_O)_7_(DMF)_5.5_, which was determined by elemental analysis (CCDC 2042391 contains the supplementary crystallographic data for this paper. This data can be obtained free of charge from The Cambridge Crystallographic Data Centre via www.ccdc.cam.ac.uk/data_request/cif (accessed on 4 November 2020)). The configuration of the Zn centers were classified as three groups in terms of their coordination atoms: (1) Zn1 was coordinated with six O atoms from six independent ligands; (2) Zn2 was coordinated with six O atoms from four independent ligands; (3) Zn3 was coordinated with five O atoms from four independent ligands (Figure 1a). It is worth noting that the ratio of Zn1:Zn2:Zn3 was 1:1:1. As for the coordination patterns of the ligand, every ligand had four carboxyl groups and were coordinated to seven metal ions in total (Figure 1b). In addition, the frameworks had a one-dimensional pore along the *c*-axis with a size of 3.41 × 5.17 Å^2^ (Figure 1c), which was theoretically exactly suitable to encapsulate the DPEE and DPEM molecules (Appendix A). Figure 1d presents a simplified topology, where blue and red represent the two independent interpenetrated frameworks.

The PXRD patterns of the as-synthesized Zn-CYMPN and Zn-CYMPN⊃dyes matched well with the simulated ones obtained from the single-crystal X-ray diffraction analysis, validating that the Zn-CYMPN was synthesized successfully and that the encapsulation of the dyes did not destroy the structure of the MOFs (Appendix A). The binding of the guest molecules and the MOFs was only due to the host–guest interaction [30,31]. The results of the Brunauer–Emmett–Teller (BET) test are shown in Figure 2. The specific surface areas of the Zn-CYMPN and Zn-CYMPN⊃DPEE-1 were 27.4345 m^2^/g and 8.9918 m^2^/g, respectively. Although the value was small, the specific surface areas of the MOFs still decreased after the encapsulation of the dyes, certifying that the dyes were loaded into the pores of the Zn-CYMPN. The content of dyes in the Zn-CYMPN–dye composites were determined by ultraviolet–visible absorption spectra (Appendix A), which are shown in Table 1.

### 2.2. Thermostability

The thermostability of the dyes, Zn-CYMPN and Zn-CYMPN⊃dyes were detected by thermogravimetric analyses (TGA) in a N_2_ atmosphere. For easier comparison, all the samples were tested under the same conditions. The analysis of their thermogravimetric curves showed that the weight loss processes of the Zn-CYMPN and Zn-CYMPN⊃dyes could be roughly divided into two stages, namely 40~150 °C and 400~450 °C, respectively (Appendix A). The first stage was mainly caused by the loss of solvent molecules such as DMF, water, etc. The second stage resulted from the decomposition of the organic ligands of the frameworks. From the TG curves of the DPEE and DPEM, we found that the two dyes both lost weight quickly at 250 °C due to the volatilization or decomposition of the molecules, and the weight-loss process ended at 400 °C, maintaining 10% of the original weight up to 800 °C. However, on the thermogravimetric curves of the Zn-CYMPN⊃DPEE and Zn-CYMPN⊃DPEM, there was no rapid weight loss at 250 °C, and the loss started from 400 °C, demonstrating that the encapsulation could enhance the thermostability of the dyes. The Zn-CYMPN⊃dye materials were well-suited for use in wide temperature regions for fluorescent temperature detection. This also indicated that the DPEE and DPEM in the Zn-CYMPN⊃dyes were evenly encapsulated into the pores of the Zn-CYMPN instead of in their crystalline state, which may reduce the ACQ of the dye molecules and also improve the quantum yields enormously.

### 2.3. Quantum Yields

The quantum yields (QY) of the DPEE, DPEM, Zn-CYMPN⊃DPEE and Zn-CYMPN⊃DPEM were measured to testify whether the ACQ of the dye molecules was reduced. As shown in Table 1, the fluorescent quantum yields of the Zn-CYMPN⊃DPEE and Zn-CYMPN⊃DPEM were higher than that of their corresponding dye powders, indicating that encapsulation by Zn-CYMPN effectively reduced the ACQ and improved the quantum yield of the dyes.

### 2.4. Fluorescent Properties

In order to determine the fluorescent properties of the Zn-CYMPN⊃dye composites, The excitation and emission spectra of the DPEE, DPEM, Zn-CYMPN, Zn-CYMPN⊃DPEE, and Zn-CYMPN⊃DPEM at room temperature were examined (Appendix A). Upon excitation at 466 nm, Zn-CYMPN⊃DPEE exhibited a strong characteristic emission of DPEE at about 620 nm (Appendix A); Zn-CYMPN⊃DPEM also exhibited a strong characteristic emission of DPEM at about 615 nm under the same excitation light (Appendix A). The fluorescent emission of Zn-CYMPN⊃DPEE and Zn-CYMPN⊃DPEM were similar to the corresponding dyes dispersed in DMF (Appendix A). These results demonstrated that the Zn-CYMPN could provide a suitable environment for the fluorescent emissions of DPEE and DPEM. With their high temperature stability, relatively high quantum yields and excellent fluorescent properties, Zn-CYMPN⊃DPEE and Zn-CYMPN⊃DPEM can be used as fluorescent temperature-detecting materials in a wide temperature region.

### 2.5. Fluorescent Temperature Detection

To assess the potential of the Zn-CYMPN⊃dyes as multi-parameter thermometers, the temperature-dependent emission spectra of the Zn-CYMPN⊃DPEE-1 were recorded in the temperature range from 40 °C to 250 °C (Figure 3a). As the temperature rose, the fluorescence intensity dropped because of heat quenching. Specifically, with the increase in temperature, the non-radiative transition of the dye molecules was enhanced, and the luminescence was weakened. Since the fluorescent intensity at 616 nm showed a good linear relationship with the temperature in the range from 50 °C to 200 °C, it can be used as a parameter for temperature detection, thereby realizing the detection in a relatively wide temperature region (Figure 3b). The fitting function is as follows:*I* = −0.0055*T* + 1.1397(1)
where *I* represents the temperature-dependent parameter intensity at a specific fixed wavelength. With the correlation coefficient of *R*^2^ = 0.98, Zn-CYMPN⊃DPEE-1 is an excellent fluorescent thermometer within the temperature range of 50 °C to 200 °C.

The relative sensitivity (*S_r_*) is an important index and is commonly used to evaluate and compare the properties of different thermometers [9]. By substituting function (1) into the function of the relative sensitivity:(2)$$S_{r}=\left|\frac{\partial I/\partial T}{I}\right|$$

The relative sensitivity of Zn-CYMPN⊃DPEE-1 is obtained, which steadily increased from 0.61% K^−1^ to 6.91% K^−1^ from 50 °C to 200 °C (Figure 3c). The maximum relative sensitivity of *S_m_* = 6.91% K^−1^ was higher than most of the other available fluorescent thermometers working in this temperature range [9].

On the other hand, the maximum emission wavelength of the DPEE in the Zn-CYMPN⊃DPEE-1 (denoted as *W*) increased as the temperature rose, which could also be used as another parameter to detect temperature. The mechanism of wavelength movement has also been studied in several studies. The shift in the fluorescence emission peak position is basically related to the change in the energy band structure, and a temperature change can affect the electronic band structure of materials [32,33,34,35]. The positions of the maximum fluorescence peaks at different temperatures are shown in Figure 3d. There was also a good linear relationship between *W* and temperature in the range from 40 °C to 250 °C, which could be fitted as function of:*W* = 0.069*T* + 614.253(3)

According to the fitting function, the corresponding wavelength shifted per degree centigrade was about 0.07 nm.

Together with the fluorescence intensity, the maximum emission wavelength could make the detection results immune to power oscillations, light source fluctuations, sample background fluorescence, probe concentrations and the geometric size of the material, which means that the temperature can be determined by multiple self-referencing parameters in the Zn-CYMPN⊃DPEE-1 composite. Therefore, using both the fluorescence intensity and the maximum emission wavelength to determine the temperature could improve the accuracy of temperature measurement.

To verify the availability of the compounds, the reversibility of the Zn-CYMPN⊃DPEE-1 was further evaluated by studying the robustness of the fluorescent thermometer through three heating–cooling cycles between 40 °C and 250 °C. The results are shown in Figure 4. The emission intensity in the temperature range from 40 °C to 250 °C was reversible, indicating the excellent reliability and reusability of the sensor.

To illustrate the universality of the design method for this material system, the loading density and types of dyes were changed, and Zn-CYMPN⊃DPEE-2, Zn-CYMPN⊃DPEE-3, Zn-CYMPN⊃DPEM-1, Zn-CYMPN⊃DPEM-2 and Zn-CYMPN⊃DPEM-3 were obtained. These composites showed similar fluorescent temperature-detection performances, indicating that the system had good versatility. As shown in Appendix A, there were good linear relationships between *I* or *W* and temperature, which could be fitted as functions of:(4)I=−0.0053T+1.1351 (Zn-CYMPN⊃DPEE-2)
(5)W=0.083T+605.512 (Zn-CYMPN⊃DPEE-2)
(6)I=−0.0050T+1.0393 (Zn-CYMPN⊃DPEE-3)
(7)W=0.057T+619.213 (Zn-CYMPN⊃DPEE-3)
(8)I=−0.0030T+0.7210 (Zn-CYMPN⊃DPEM-1)
(9)W=0.060T+615.301 (Zn-CYMPN⊃DPEM-1)
(10)I=−0.0031T+0.7685 (Zn-CYMPN⊃DPEM-2)
(11)W=0.068T+610.072 (Zn-CYMPN⊃DPEM-2)
(12)I=−0.0032T+0.7971 (Zn-CYMPN⊃DPEM-3)
(13)W=0.065T+609.550 (Zn-CYMPN⊃DPEM-3)

## 3. Materials and Methods

All solvents and reagents were commercially available and used without further purification. Powder X-ray diffraction (PXRD) data were collected within the range of 2θ = 3°~50° on a X’Pert Pro X-ray diffractometer using a Cu-*K*_α_ (λ = 1.542 Å) beam at room temperature. Single-crystal X-ray diffraction data for Zn-CYMPN were recorded on a Bruker SMART APEX II diffractometer. Thermogravimetric analyses (TGA) were conducted on a Netzsch TGA 209 F3 thermogravimeter with a heating rate of 10 K·min^−1^ under a N_2_ atmosphere from 40 °C to 800 °C. Room-temperature emission and excitation spectra for the samples were detected by using a Hitachi F4600 fluorescence spectrometer. The excitation light source was a xenon lamp with excitation and emission slit sizes of 5 nm and 2.5 nm, respectively. The PMT detector’s voltage depended on the samples, and the scanning speed was 1200 nm/min. A temperature changer was used to control the temperature when testing the temperature-dependent emission spectra. Quantum yields of the samples were obtained on an Edinburgh FLS920 multi-function fluorescence spectrometer in an absolute method with a xenon lamp as the excitation light source and red PMT as a detector. Ultraviolet–visible absorption spectra were obtained on a U4100 spectrometer. Elemental analysis (EA) was conducted on a Flash EA1112 elemental analyser. The data of the BET test were collected on a MicromeriticsASAP2020 gas adsorption instrument, and the samples needed to be activated before testing.

Synthesis of Zn-CYMPN: Zn-CYMPN was synthesized following the steps below: H_6_CYMPN (4.2 mg, 0.00514 mmol) and Zn(NO_3_)_2_·6H_2_O (1.53 mg, 0.00514 mmol) were dissolved in 1.4 mL of dimethylformamide (DMF) and 0.6 mL H_2_O in a reaction vessel. Then, these reaction vessels were placed in an oven at 110 °C for two days. The resulting colourless crystals were washed three times with DMF and ethanol. Then, they were dried in an oven at 60 °C. Optical photographs of the MOFs with excitation at 380 nm using a mercury lamp can be found in Appendix A.

Synthesis of Zn-CYMPN⊃dyes: the crystals of Zn-CYMPN were immersed in DMF solutions of DPEM or DPEE with three different concentrations of 1 × 10^−3^, 1 × 10^−2^ or 1 × 10^−1^ mol/L for four days. The products were washed three times with DMF to remove residual dyes. Then, they were dried in an oven at 60 °C. The names of the samples are shown in Table 1. An optical photograph is provided in Appendix A with a 380 nm excitation.

## 4. Conclusions

In conclusion, a new type of fluorescent thermometer that encapsulated dyes in the pores of Zn-CYMPN was designed and synthesized. The resulting MOF⊃dye composites had a fairly uniform dye distribution and excellent thermal stability, which effectively increased the quantum yield by preventing the self-quenching of the dyes caused by ACQ. The thermometers Zn-CYMPN⊃DPEE and Zn-CYMPN⊃DPEM had excellent fluorescent properties, enabling a wide range of temperature detection. The compounds could measure the temperature using multiple parameters (fluorescence intensity and maximum emission wavelength), which greatly improved the reliability of temperature detection, and the universality of this MOF⊃dye system was proven by changing the kind and loading density of the dyes. This work will provide some ideas for the study of MOF⊃fluorescent guest species composite materials, and the wide selection of guests as well as the structural reliability of the MOFs not only allows the design of fluorescent sensors with multi-parameter readings but also affords the potential of more diverse emission ranges and higher sensitivities.

## Figures and Tables

**Figure 1 molecules-28-00729-f001:**
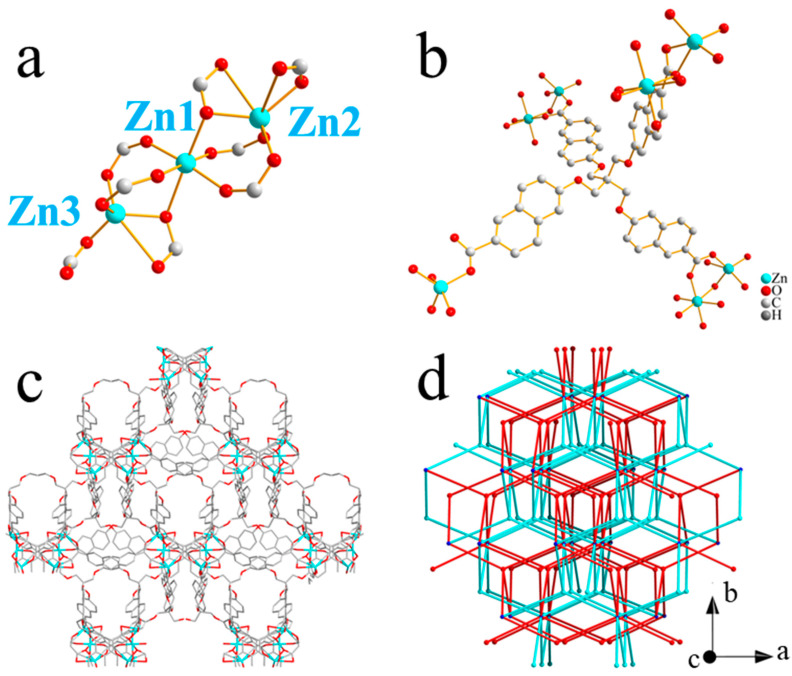
Coordination environment of (**a**) Zn and (**b**) ligand. (**c**) one-dimensional channel along *c*-axis with size of 3.41 × 5.17 Å^2^. (**d**) Simplified topology; blue and red represent two independent interpenetrated frameworks.

**Figure 2 molecules-28-00729-f002:**
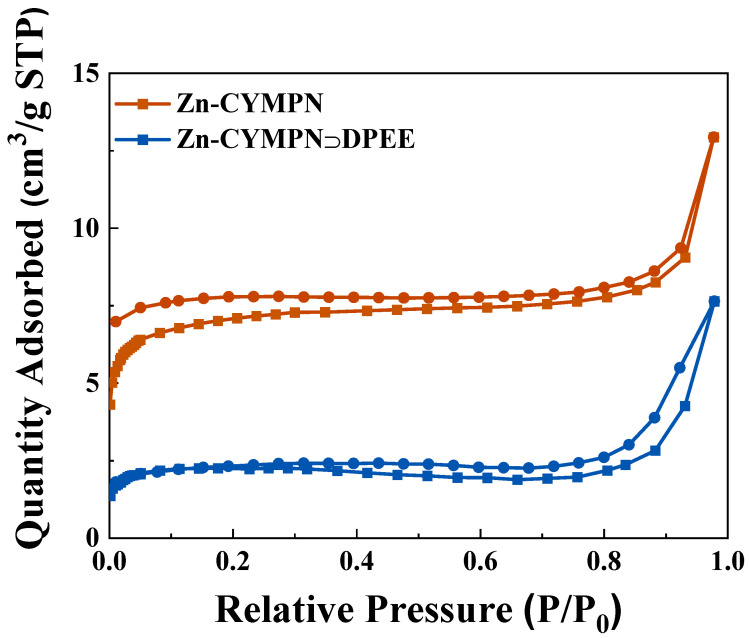
The N_2_ adsorption/desorption isotherms of Zn-CYMPN and Zn-CYMPN⊃DPEE. Square symbols: adsorption; circular symbols: desorption.

**Figure 3 molecules-28-00729-f003:**
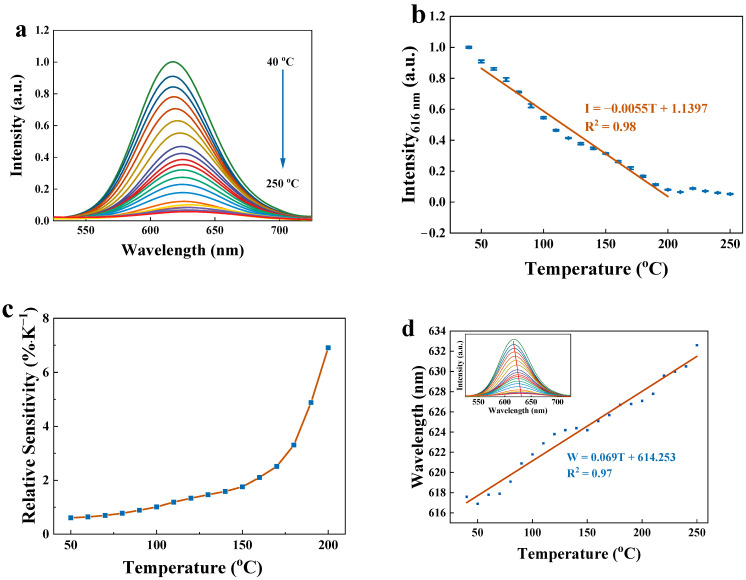
(**a**) Temperature-dependent fluorescent emission spectra of Zn-CYMPN⊃DPEE-1. (**b**) Fluorescence intensity at 616 nm, and the fitting line with temperature of 50~200 °C. (**c**) The relative sensitivity for intensity. (**d**) Wavelength at the maximum fluorescent emission and the fitting line with temperature of 40~250 °C (insert: temperature-dependent fluorescent emission spectra and the shift in wavelength corresponding to the maximum fluorescence emission intensity).

**Figure 4 molecules-28-00729-f004:**
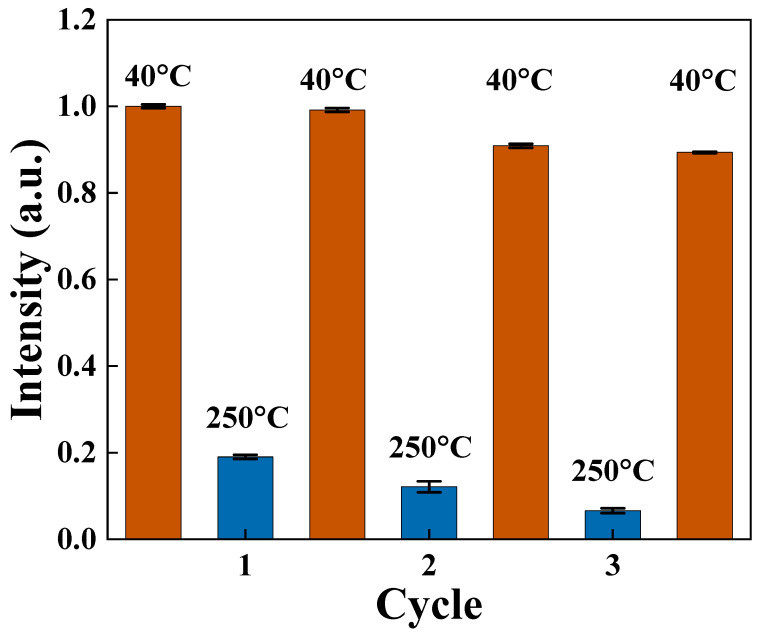
The relationship between fluorescence intensity at 616 nm of Zn-CYMPN⊃DPEE-1 and temperature during three thermocycles in the range of 40~250 °C.

**Table 1 molecules-28-00729-t001:** The name, loading density and quantum yield of the samples.

Sample	Concentration of DPEE (mol/L)	Concentration of DPEM (mol/L)	Loading Density of DPEE	Loading Density of DPEM	Quantum Yield
Zn-CYMPN⊃DPEE-1	1 × 10^−1^	/	4.20 wt%	/	17.87%
Zn-CYMPN⊃DPEE-2	1 × 10^−2^	/	0.63 wt%	/	16.71%
Zn-CYMPN⊃DPEE-3	1 × 10^−3^	/	0.26 wt%	/	3.20%
Zn-CYMPN⊃DPEM-1	/	1 × 10^−1^	/	3.40 wt%	30.40%
Zn-CYMPN⊃DPEM-2	/	1 × 10^−2^	/	3.07 wt%	5.45%
Zn-CYMPN⊃DPEM-3	/	1 × 10^−3^	/	2.91 wt%	2.47%
DPEE	/	/	/	/	1.19%
DPEM	/	/	/	/	2.13%

## Data Availability

Not applicable.

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
