# Peer review of "Dye-Encapsulated Metal–Organic Frameworks for the Multi-Parameter Detection of Temperature"

_molecules, 2023, doi:10.3390/molecules28020729_

Round 1
Reviewer 1 Report
1. Introduction. Explain the choice of H6CYMPN as a linker
2. Please check if all abbreviations in the text are explained
3. The Fig. 1d is not referred in the main text.
4. As a good sensing materials, it should be tested the stability, I suggest the authors to check the PXRD after the samples coating in different solvents for more days.
5. Since two new compounds have been synthesized, I suggest the authors to delete the “novel” in the full text.
6. The authors have measure the Quantum yields, but I cannot find any figures in the main text?
7. Please add brief section about fluorescent sensing materials, please read Mater. Today. Commum., 2022, 31,103514;Coord. Chem. Rev., 445(2021) 214074; Catal. Sci. Technol., 2021, 11, 3946–3989 and Coord. Chem. Rev. 2020, 406:213145
8. Please provide a full 3D network in the Figure 1.
9. Compare the similar work and highlight them
10. Please provide the naked photos before and after encapsulated the dyes.
Author Response
Thank you for the patient reading. Our detailed responses are as follows:
- Introduction. Explain the choice of H6CYMPN as a linker
Response: There are two main considerations when we choose H6CYMPN as a linker. First of all, the ligand is a new type of structure, so it is hopeful to obtain a new MOF material, which is innovative. In addition, the longer side chains of H6CYMPN may interpenetrate with each other, which will form smaller channels, helping the guest molecules to bind more firmly. Relevant description has been added to the introduction section.
- Please check if all abbreviations in the text are explained.
Response: We carefully checked the article to make sure that all abbreviations are explained.
- The Fig. 1d is not referred in the main text.
Response: Thank the reviewer for pointing out this detail. Relevant description has been added.
- As a good sensing materials, it should be tested the stability, I suggest the authors to check the PXRD after the samples coating in different solvents for more days.
Response: The original intention of our design is to use solids for sensing to avoid the disadvantages such as volatility of liquid dye molecules, so stability tests in the liquid environment were not carried out. In addition, due to the possible structural instability of the material caused by the longer side chain of the ligand, the solvent stability of the composite may be not as good as expected.
- Since two new compounds have been synthesized, I suggest the authors to delete the “novel” in the full text.
Response: Thank the reviewer for pointing out this key issue. Due to the novelty of the ligand, a novel structure Zn-CYMPN is generated. We tend to believe that this MOF has the novelty of structure and composition. However, we have reduced the use of this word.
- The authors have measure the Quantum yields, but I cannot find any figures in the main text?
Response: Experiment results about Quantum yields are presented in the text in the form of Table 1.
- Please add brief section about fluorescent sensing materials, please read Mater. Today. Commum., 2022, 31, 103514; Coord. Chem. Rev., 445 (2021) 214074; Catal. Sci. Technol., 2021, 11, 3946–3989 and Coord. Chem. Rev. 2020, 406: 213145
Response: The brief introduction of fluorescent sensing materials has been added. Thanks to the reviewer for the literature recommendation. We have read the relevant review in detail and cited it in this article.
- Please provide a full 3D network in the Figure 1.
Response: The full 3D network of Zn-CYMPN has been provide as Figure 1d.
- Compare the similar work and highlight them
Response: In Introduction, several optical sensing references have been cited, and also we introduced Professor Qian's work on the implementation of multimode sensing by ZIF-8, and relevant introductions are also supplemented in the manuscript.
- Please provide the naked photos before and after encapsulated the dyes.
Response: The photos before and after encapsulated the dyes are provided in Supplementary as Figure S13. In order to display more clearly, we choose ultraviolet light excitation to obtain the luminous image of the material.
Reviewer 2 Report
1. It is important to evaluate the thermal stability of the two compounds. However, in the manuscript, the author listed the test instruments and found no relevant content. Why ?
2. In the assembly of the crystal materials,the influence of host/guest ratios on the structure of compounds is important. In this regard, the author should discuss it and some related literature should be cited, Inorg. Chem. 2021, 60, 18593-18597; Cryst. Growth Des. 2022, 22, 4018-4024;
3. Finally, there are some many tedious sentences, and the paper should be amended comprehensively.
4. Topological analysis might need to be repeated for further simplification of nets. Cluster motifs (containing several Zn atoms) can be considered as nodes and this can lead to simplified topologies.
5. Some subfigures are too small. Increase size/resolution or change layout.
6. Introduction. When talking about the role of organic ligands for assembling functional coordination polymers, such as Inorg. Chem., 2022, 61, 9328-9338; Chem. Soc. Rev., 2022, 51, 6126-6176; CrystEngComm, 2022, 24, 7157–7165; New J. Chem., 2022, 46, 19577–19592 and Dalton Transactions, 2022, 51, 14817-14832.
Author Response
Thank you for the patient reading. Our detailed responses are as follows:
- It is important to evaluate the thermal stability of the two compounds. However, in the manuscript, the author listed the test instruments and found no relevant content. Why?
Response: Thank the reviewers for their careful reading. The relevant content was provided in Supplementary as Figure S4, and part 2.2 Thermostability in Results and Discussion section.
- In the assembly of the crystal materials, the influence of host/guest ratios on the structure of compounds is important. In this regard, the author should discuss it and some related literature should be cited, Inorg. Chem. 2021, 60, 18593-18597; Cryst. Growth Des. 2022, 22, 4018-4024;
Response: Thanks for the suggestions of the reviewer. As mentioned by the reviewer, the proportion of host and guest will affect the final material performance. Therefore, we have designed several experimental groups with different dye loading contents. Relevant description and literature have been added.
- Finally, there are some many tedious sentences, and the paper should be amended comprehensively.
Response: On the expression of the article, we revised the full text and have tried our best to make some changes.
- Topological analysis might need to be repeated for further simplification of nets. Cluster motifs (containing several Zn atoms) can be considered as nodes and this can lead to simplified topologies.
Response: Topological analysis were repeated to simplify the net. As for the further simplification of nets and clearer topology, we have modified Figure 1d, hoping to express the 3D structure more clearly.
- Some subfigures are too small. Increase size/resolution or change layout.
Response: Thanks for the professional opinions of the reviewers, we have made changes to all subfigures.
- Introduction. When talking about the role of organic ligands for assembling functional coordination polymers, such as Inorg. Chem., 2022, 61, 9328-9338; Chem. Soc. Rev., 2022, 51, 6126-6176; CrystEngComm, 2022, 24, 7157–7165; New J. Chem., 2022, 46, 19577–19592 and Dalton Transactions, 2022, 51, 14817-14832.
Response: We have carefully read the above references, and thank the reviewers for their recommendations.
Reviewer 3 Report
A series of dyes encapsulated MOFs Zn-CYMPN-DPEE and Zn-CYMPN-DPEM were synthesized as fluorescent thermometers, which show change in both fluorescent intensity and wavelength along with increasing temperature. The immobilization of dyes and maintaining of crystallinity were confirmed by XRD and N2 adsorption. This work provides a new avenue for MOF based fluorescent temperature sensors yet interpretation of some data is mistaken. Thus, I suggest the work to be accepted with major revision.
Comments:
1. The BET surface area was reported with major decrease after guest introduction. Yet the pore size distribution should also be provided to see which portion of pores were blocked.
2. The author claimed a multiple-parameter thermometer with changes on both intensity and wavelength over temperature change. However, change in wavelength with intensity is no new finding which is due to the band structure moving. Also, from Figure 3d we could see R2=0.97, which is not reliable enough as a sensing parameter. General experimental errors could affect the result.
3.For intensity change over temperature, two separate trend can be observed for 50-100 and 100-200 degree. This trend is clearer in Figure S9b. The reason for R2 as low as 0.95 is because that the data should be processed in two section: 50-100 and 100-200degree. This often happens when structure or phase of the material changes through stimulation. The author should do deeper literature review and understand this phenomenon.
4. The error bar in intensity is observed in supporting information but not main text. Error bar and standard deviation should be given as well.
Author Response
Thank you for the patient reading. Our detailed responses are as follows:
- The BET surface area was reported with major decrease after guest introduction. Yet the pore size distribution should also be provided to see which portion of pores were blocked.
Response: Thank reviewers for their professional questions. For MOFs system, whether before or after dye loading, the pore size of our MOF is relatively small. Therefore, based on the unchanging single crystal XRD analysis results before and after loading and compared to the size of dyes, it is a reasonable deduce that the dye exists in the largest channel (3.41×5.17 Å2) shown in Figure 1c.
- The author claimed a multiple-parameter thermometer with changes on both intensity and wavelength over temperature change. However, change in wavelength with intensity is no new finding which is due to the band structure moving. Also, from Figure 3d we could see R2=0.97, which is not reliable enough as a sensing parameter. General experimental errors could affect the result.
Response: For the physical phenomenon of wavelength shift, it is indeed not a new discovery. However, we combine novel MOFs, fluorescent dye, and temperature sensing with the above physical phenomena, hoping to provide researchers some new inspiration more or less. We know that 0.97 for the linear fitting here is not particularly excellent, but there is no doubt that linear fitting is the most intuitive one of all fitting methods, which is helpful for subsequent measurement. In addition, error bars were added to all data points already, which can provide reference for readers to a certain extent.
3.For intensity change over temperature, two separate trend can be observed for 50-100 and 100-200 degree. This trend is clearer in Figure S9b. The reason for R2 as low as 0.95 is because that the data should be processed in two section: 50-100 and 100-200 degree. This often happens when structure or phase of the material changes through stimulation. The author should do deeper literature review and understand this phenomenon.
Response: Thank reviewers for their professional suggestions. Indeed, structural changes will lead to changes in optical properties. Ideally, we should obtain PXRD data of materials at high temperatures to verify this assumption, and we will study deeper in the future. As for the piecewise fitting problem you mentioned, we considered this method when sorting out the data, but in order to simplify the detection method, we finally gave up this relatively troublesome method. Thanks to the reviewers for the inspiration again.
- The error bar in intensity is observed in supporting information but not main text. Error bar and standard deviation should be given as well.
Response: In fact, the error bars in the main text is also given, but the display here is not very obvious because of its small value. We have adjusted the format and display of the figures in the full text, hoping to present a better effect.
Round 2
Reviewer 3 Report
Most of the questions were answered yet I still believe that the fitting in Figure S9b should be more scientific and really follow the trend. To simplify the testing process is not a good reason for low R2 coefficient. The author raised a good suggestion about using PXRD to verify the structure change at high temperature. Hope this could be answered in the future work.
Thus, I suggest the work to be accepted after minor revision.
Author Response
Thank you for providing the good evaluation of this article on the whole and the inspiration for the future work, but there is one problem we still want to adhere to the original author's data processing method. We do understand that reviewers have very strict standards for data fitting, but as we explained the reasons in our last reply, in order to maintain the uniformity of fitting methods in the whole and to meet the needs of subsequent probe production, we still want to adopt a consistent fitting method, that is, a relatively simple linear fitting, even if the fitting accuracy will be reduced to a certain extent.
In fact, when considering the fitting method, we first considered the possible physical significance of the fitting formula. But unfortunately, this article finally chose to emphasize the use of two signals for temperature sensing, rather than more in-depth analysis of the physical mechanism. And of course, we will continue to go deep into this work in the future, and we will also learn from your opinions and consider other fitting methods.